# The Extraction of β-Carotene from Microalgae for Testing Their Health Benefits

**DOI:** 10.3390/foods11040502

**Published:** 2022-02-10

**Authors:** Jing Wang, Xinge Hu, Junbin Chen, Tiannan Wang, Xianju Huang, Guoxun Chen

**Affiliations:** 1College of Pharmacy, South-Central University for Nationalities, Wuhan 430074, China; 18772842249@163.com (J.W.); xianju@mail.scuec.edu.cn (X.H.); 2Department of Nutrition, University of Tennessee at Knoxville, Knoxville, TN 37996, USA; xhu25@vols.utk.edu (X.H.); twang14@vols.utk.edu (T.W.); 3School of Public Health, Southern Medical University, Guangzhou 510515, China; box@i.smu.edu.cn

**Keywords:** microalgae, β-carotene, carotenoids, vitamin A, extraction, bioactivities

## Abstract

β-carotene, a member of the carotenoid family, is a provitamin A, and can be converted into vitamin A (retinol), which plays essential roles in the regulation of physiological functions in animal bodies. Microalgae synthesize a variety of carotenoids including β-carotene and are a rich source of natural β-carotene. This has attracted the attention of researchers in academia and the biotech industry. Methods to enrich or purify β-carotene from microalgae have been investigated, and experiments to understand the biological functions of microalgae products containing β-carotene have been conducted. To better understand the use of microalgae to produce β-carotene and other carotenoids, we have searched PubMed in August 2021 for the recent studies that are focused on microalgae carotenoid content, the extraction methods to produce β-carotene from microalgae, and the bioactivities of β-carotene from microalgae. Articles published in peer-reviewed scientific journals were identified, screened, and summarized here. So far, various types and amounts of carotenoids have been identified and extracted in different types of microalgae. Diverse methods have been developed overtime to extract β-carotene efficiently and practically from microalgae for mass production. It appears that methods have been developed to simplify the steps and extract β-carotene directly and efficiently. Multiple studies have shown that extracts or whole organism of microalgae containing β-carotene have activities to promote lifespan in lab animals and reduce oxidative stress in culture cells, etc. Nevertheless, more studies are warranted to study the health benefits and functional mechanisms of β-carotene in these microalgae extracts, which may benefit human and animal health in the future.

## 1. Introduction

Microalgae are a large group of photosynthetic microorganisms including unicellular prokaryotic and eukaryotic organisms [1]. They are widely distributed in land and sea, and are rich sources of nutrients [2,3]. The size of microalgae ranges from 0.2 to 2 μm (picoplankton) up to 100 μm or higher (filamentous forms) [4]. They are mainly autotrophic, but a few microalgae are heterotrophic due to the degeneration of chloroplasts [5]. Microalgae perform photosynthesis to produce organic molecules and support their rapid growth. Their short mitotic time allows them to be produced on a large scale for the extraction of bioactive compounds, which have a lot of biological activities in functional food and nutraceuticals, as summarized in an edited book [6]. Microalgae exist in diverse environmental conditions and are rich sources of biomolecules such as proteins, fat, and carbohydrates. Microalgae culture can be industrialized to produce compounds with commercial value [7,8]. As microalgae synthesize lipids, carbohydrates, proteins, vitamins, and pigments, they are a food source for plankton [9]. Microalgae are also considered a candidate to fix carbon dioxide for biofuel production, and to sequestrate nitrogen oxides and sulfur oxides for sewage treatment and environmental protection [10]. Due to the immature technology and high production cost, microalgae biofuel has not been industrialized [11]. On the other hand, outbreaks of harmful or toxic microalgae can cause red tides and algal blooms in eutrophication [12].

Recently, microalgae have attracted intensive research interests due to their commercial potential [13]. The diverse bioproducts from microalgae can be widely used for pharmaceuticals, nutraceuticals, food colorants, and animal feed. At present, microalgae in the following phyla: *Cyanophyta*, *Chlorophyta*, *Chrysophyta*, and *Erythrophyta* have been cultivated or produced in large quantities. The common microalgae with economic values include *Haematococcuspluvialis*, *chlorella*, and *spirulina* [14,15]. Compared with the higher plants, microalgae present a series of advantages such as faster growth, higher yield, and shorter cultivation time. Therefore, they are widely used in the industrial production of bioactive compounds [16]. It has been estimated that more than 200 bioactive compounds can be extracted from cyanobacteria, and thousands from eukaryotic microalgae [16]. For example, marine organisms accumulate vitamin D through the consumption of (micro)algae [17]. In addition, *Spirulina* is also called a “superfood” because of its contents of vitamins (E, K, B_1_, B_2_, B_3_, B_6_, and B_12_), proteins (phycocyanin, allophycocyanin, and phycoerythrin) and other bioactive compounds (γ-linolenic acid, palmitic acid, calcium, selenium, zinc, etc.) [18,19].

Microalgae are one source of carotenoids. *Spirulina*, *Chlorella*, *Dunaliella*, and *Haematococcus* can produce fucoxanthin, violaxanthin, neoxanthin, α-carotene, β-carotene, and lutein in large quantities, which are considered provitamin A [20]. β-carotene, a ubiquitous pigment concerned with the photosynthetic process in microalgae, has shown a variety of bioactivities [21]. It is the most abundant dietary provitamin A that can be converted into vitamin A (VA, retinol), which is a micronutrient for human health [22]. The raw extract and pure compound of β-carotene from microalgae have been studied and shown to have biological activities such as hepatotoprotection, anti-obesity, anti-inflammation, immunomodulation, and anti-cancer [23,24,25,26]. For example, supplementations of β-carotene extracts from *Spirulina* and *Dunaliella* have been shown to reduce the activity of transaminases in CCl4-induced hepatic-damaged Wistar rats [27]. *Spirulina* is rich in carotenoids, and the supplementation of *Spirulina* biomass raises the antioxidant enzymes in the serum and liver of Wistar rats, showing its antioxidant activity [28].

To date, various methods such as solvent extraction and supercritical fluid extraction have been developed to extract carotenoids from microalgae for mass production [29]. Despite previous concerns such as efficiency, high solvent-consumption, and long treatment times [30], methods have been evolved to solve these problems. For example, these methods have been used to extract carotenoids from vegetables [31]. It was estimated that about 50% of total β-carotene in *Scenedesmus almeriensis* can be extracted using a supercritical carbon dioxide extraction method [32]. The aim of this review is to summarize β-carotene, its extraction methods, its health value, and the presence of other types of carotenoids in different microalgae sources. In addition, we wanted to evaluate the studies that investigated bioactivities of β-carotene obtained from microalgae. Therefore, we searched the PubMed database in August 2021, retrieved relevant peer-reviewed articles that discussed the extraction methods of β-carotene, and evaluated the bioactivities of extracts containing β-carotene and other carotenoids from microalgae in cell and animal models. We hope to identify gaps for the future use of β-carotene and other carotenoids in the promotion of human health.

## 2. Vitamin A (VA) and β-Carotene

### 2.1. VA and Its Metabolism

VA (retinol) is essential for the general health of humans [33]. Molecules with VA activities include preformed VA and provitamin A. Preformed VA in the forms of retinyl esters (REs) and retinol can be found in foods from animal products [22]. Liver and fish oil have the highest concentration of preformed VA provitamin A carotenoids, mainly from plants and microalgae [33]. The most important provitamin A carotenoid is β-carotene, which is present abundantly in carrots and yellow and green leafy vegetables [33]. Other provitamin A carotenoids include α-carotene and β-cryptoxanthin. Carotenoids are also solubilized into micelles in the intestinal lumen from which they are absorbed into duodenal mucosal cells [22]. Carotenoids (such as β-carotene) after absorption can be converted into retinal within the enterocytes or absorbed and transported to hepatocytes for cleavage into retinal (retinaldehyde), which can be reduced into retinol [34].

In the small intestine lumen, dietary REs are hydrolyzed into free retinol and fatty acids by the intraluminal retinol ester hydrolases [34] and nonspecific pancreatic enzymes such as pancreatic triglyceride lipase and cholesterol ester hydrolase. Retinol and fatty acids are absorbed into enterocytes [35], where they are esterified mainly by lecithin:retinol acyltransferase to form REs and packed into chylomicrons. REs and carotenoids in chylomicrons can be taken up directly by peripheral tissues [22]. The remaining REs and carotenoids in chylomicron remnants are taken by the liver [22].

The plasma retinol concentration is under homeostatic control, which reflects the dietary VA intake and hepatic VA reserve [36]. To meet the tissue needs for VA, retinol released from the liver binds to retinol binding protein 4 (RBP4), which is synthesized in hepatocytes and responsible for VA transport in the body [22]. Retinol binds to RBP4 in the blood, and cellular retinol binding proteins in cells [37,38]. Retinol can be oxidized by retinol dehydrogenase to retinal (retinaldehyde), which can be reduced by alcohol dehydrogenases/retinol dehydrogenases to retinol again or oxidized by retinaldehyde dehydrogenase into retinoic acid (RA). RA enters into the cell nucleus and regulates gene expressions through activations of transcription factors such as RA receptors and retinoid X receptors, which control cell morphogenesis, differentiation, proliferation, etc. [34]. Further oxidization of RA mediated by Cyp26A1 and Cyp26B1 generates polar compounds without the ability to activate transcription. Retinol can be esterified by lecithin:retinol acyltransferase into REs for storage [39,40]. When the liver VA content rises, the excretion of its metabolites in the bile increases. Other VA metabolites are excreted in the urine [22]. In addition, VA can act as an antioxidant to reduce free radicals [41].

In 2001, the American Institute of Medicine defined the Recommended Dietary Allowance (RDA) of VA in micrograms (μg) of retinol activity equivalent (RAE) to illustrate the different biological activities of retinol and provitamin A carotenoids. One µg of RAE is equivalent to 1 µg of retinol, 2 µg of supplementary β-carotene, 12 µg of dietary β-carotene, or 24 µg of dietary α-carotene or β-cryptoxanthin [33]. The RDAs are 900 μg RAE for men, 700 μg RAE for women, and 770 μg RAE for pregnant women aged between 19 and 50 years old [33]. Adequate VA is defined as plasma retinol levels > 1.05 μmol/L. A retinol level < 0.7 μmol/L is defined as VA deficiency.

### 2.2. β-Carotene

Carotenes are structurally different polyunsaturated hydrocarbons containing 40 carbons and synthesized by plants and microalgae. There are 1167 natural carotenoids, in which 38~50 can be considered as provitamin A, including β-carotene, β-cryptoxanthin and α-carotene, etc. [42]. The most abundant dietary provitamin A carotenoid is β-carotene, which has eight isoprene units and two β-ionone-ring at both ends [43]. Figure 1 shows the structures and some physicochemical properties of α-carotene (A), β-carotene (B), β-carotene 5,6-epoxide (C), 9-*cis*-β-carotene (D), 9-*cis*-β-carotene (E), and 9-*cis*-β-carotene (F), which can be identified in microalgae. Due to its absorbance of light, β-carotene is responsible for the color in the fungi, fruit, and vegetables such as red pepper and orange [44]. Dietary β-carotene is mainly absorbed in the duodenum portion of the small intestine, a process that is probably mediated by the class B scavenger receptor [45].

As β-carotene can be converted to retinol, its metabolism has been studied extensively. The cleavage of β-carotene symmetrically or asymmetrically leads to the production of retinal (retinaldehyde), which is reduced to retinol [46]. β-carotene cleavages primarily occur in enterocytes and hepatocytes. In the presence of oxygen, β-carotene-15,15′-oxygenase (BCO1) can symmetrically oxidize the 15,15′ double bond of β-carotene and generate two molecules of retinal (retinaldehyde). The 9, 10′ double bond of β-carotene is asymmetrically cleaved by β-carotene 9′,10′-oxygenase 2 (BCO2), which generates a β-apo-10′-carotenal. Other asymmetrical cleaves may also occur with or without enzyme catalysis and yield β-apo-8′-carotenal or β-apo-14′-carotenal, but the enzymes involved in these reactions remain unidentified [47,48]. Recent research suggested that the products of the asymmetric cleavage can be further cleaved by β-carotene-15,15′-oxygenase, which results in the production of retinal [39]. The mice with BOC1, BCO2, and BCO1/BCO2 knockout have been compared. BOC1 knockout appears to disrupt the β-carotene homeostasis, and the production of β-apo-10′ carotenol is BCO2-dependent. β-apo-10′ carotenol can be esterified and transported as retinol [49].

## 3. Microalgae as a Source of Carotenoids and Other Bioactive Compounds

### 3.1. The Applications of Carotenoids and Other Components Extracted from Microalgae

Raw extracts and pure compounds from microalgae have been studied and shown to have numerous applications and biological activities such as anti-obesity, anti-diabetes, anti-inflammation, immunomodulation, and anti-cancer [24,25,50]. For example, supplementation of *Spirulina* has been shown to reduce the body weight of overweight and obese human subjects [51,52] and increase the lipoprotein lipase activity and insulin secretion in hyperlipidemic rats [53]. *Spirulina* is rich in carotenoids, and the supplementation of *Spirulina* biomass raises antioxidant enzymes in the serum and liver of Wistar rats, showing its antioxidant activity [28].

Carotenoids have been considered as antioxidants, and they act to block the damages initiated by reactive oxygen species and maintain the integrity of cell membrane and organelles [54]. This has led people to believe that β-carotene and other carotenoids may help to reduce the risk chronic diseases, such as diabetes, cancer, and cardiovascular diseases [54].

Carotenoids including α- and β-carotene and α-tocopherol are detected in the human dermis and epidermis, as reviewed in [55]. The presence of carotenoids in the skin is thought to protect damages caused by the photooxidative processes. Carotenoids and other antioxidants can eliminate reactive oxygen species and absorb UV light, which can be achieved through dietary supplements and tropical applications [55]. In addition to β-carotene, other carotenoids can be extracted from microalgae. For example, astaxanthin can be extracted from microalga *Haematococcus pluvialis*. It has been shown that astaxanthin can decrease the oxidative stress and reduce damage caused by oxidative metabolism to skin and delay skin aging [56]. At the same time, Astaxanthin can also help repair damaged skin [57] and accelerate wound healing in mice [58]. Therefore, β-carotene and other antioxidants play important roles to maintain skin health.

Oxidized β-carotene copolymers may act as an antibacterial growth promoter to enhance the feed intake and growth in broilers [59]. The oxidized β-carotene copolymers at 2 ppm and higher supplemented in feeds improved feed conversion, average daily gain, feed intake, and body weight in tested broilers [59].

Carotenoids extracted from microalgae can be used as a pigment duo to its orange or red color. They are safer and healthier natural dyes than artificial dyes and are commonly used in the food and cosmetic industries [60]. β-carotene has been commonly added to soft drinks, cheese, and butter for coloring [61]. On the other hand, carotenoids such as astaxanthin have been used as an additive in feed to enhance the color of salmon flesh [61,62]. Canthaxanthin extracted from *Chlorella zofingiensis* is a natural dye and is commonly used as a feed additive to intensify the skin color of fish, including salmonid and crustacean [63].

Lutein is another carotenoid xanthophylls mainly extracted from *Chlorella protothecoides*. Lutein can be used to protect eyes from damage caused by short-wavelength visible light phototoxicity, thereby decreasing the risk of having cataract and age-related macular degeneration [64]. Lutein can also be used as a feed additive to improve the color of poultry egg yolks and feathers [65].

In addition to carotenoids, proteins and polyunsaturated fatty acids (PUFA) from microalgae can also have health benefits. For example, phycocyanin, a water-soluble and non-toxic protein isolated from *Spirulina*, can decrease the plasma levels of total cholesterol, triglycerides, and malondialdehyde in diabetic mice [66]. Phycocyanin exhibits anti-inflammatory activities through inhibiting cyclooxygenase-2 expression and cytokines production in lipopolysaccharide-activated macrophages [67,68]. Furthermore, microalgae are also one source of ω3 PUFAs, such as eicosapentaenoic acid (EPA) and docosahexaenoic acid (DHA). *Spirulina* and *Chlorella* are valuable sources of ω3-PUFAs [69]. To date, microalgae *Cryptocodinium cohnii* has been used to produce DHA-rich oil [70]. Microalgae of the genus *Nannochloropsis* are already exploited in aquaculture for their high content in PUFAs [70]. Studies have shown that PUFAs can reduce the incidence of cardiovascular disease [71].

### 3.2. Studies on the Carotenoid Content in Microalgae and Bioactivities

The carotenoids in microalgae include astaxanthin, β-carotene, canthaxanthin, zeaxanthin, purple xanthin, lutein, fucoxanthin, etc. β-carotene and zeaxanthin are widely distributed. High levels of β-carotene are present in *Dunaliella salina* [72,73]. The content of carotenoids in microalgae is high, which usually account for 0.1–0.2% of the total dry matter of microalgae [74]. *Dunaliella salina* produces up to 13% of its biomass as β-carotene. The content of astaxanthin was up to 7% of the biomass in *Haematococcus pluvialis*, and nearly 5% of the biomass in *Coalstrella striolata* [14]. Table 1 shows the contents of α-carotene, β-carotene, β-carotene 5,6-epoxide, 9-*cis*-β-carotene, 13-*cis*-β-carotene, and 15-*cis*-β-carotene that are identified in some microalgae. All the studies used high-performance liquid chromatography (HPLC) to analyze the presence of the indicated α-carotene, β-carotene, and its derivative or isomers. Some of them used mass spectrometry to further identify them [42,75,76,77].

The initial research on using microalgae as a potential commercial source of β-carotene began in the 1960s [82]. Later on, microalgae have been considered as a commercial source of glycerol [83]. So far, chemically synthesized carotenoids are the major products, which are about 97–98% of the total market [84]. The synthetic β-carotene products contain only all-*trans* isomer [85]. Molecules with β-carotene activities in microalgae are mixture of all-*trans* (β-carotene in this manuscript) and *cis* isomers. The *cis* isomers have stronger antioxidant capacities than the all-*trans* one. Table 1 summarizes the contents of α-carotene, β-carotene, and its derivatives, β-carotene 5,6-epoxide, 9-cis-β-carotene, 9-cis-β-carotene, and 9-cis-β-carotene in microalgae. Obtaining carotenoids from natural raw materials is favored by more and more consumers. The production of natural carotenoids to increase product values has become the new direction for researchers. Companies in Australia, Israel, and the United States of America have started to produce β-carotene from microalgae [85]. In the 1980s, β-carotene extracted from microalgae and dried powders of microalgae rich in β-carotene were marketed by companies in the United States of America and Australia [86]. The products have been used as coloring for natural foods and animal feeds, and natural β-carotene supplementation [62]. In addition to α- and β-carotene, other carotenoids are also detected in various microalgae, as shown in Table 2. HPLC is used to analyze the presence of those carotenoids. Mass spectrometry is also used in some studies shown in Table 2 [42,75,76,77]. As we can see, the contents of carotenoids such as lutein, zeaxanthin and antheraxanthin are very high in certain microalgae sources. Although more than 40,000 microalgae species have been identified, only a few of them have been used in commercial microalgae production for obtaining carotenoids and proteins, showing the future potential expansion of the field [9].

## 4. Industrial Extraction of β-Carotene from Microalgae

The production of carotenoids from microalgae includes microalgae culture, harvesting, extraction, and purification. To obtain carotenoids from microalgae, the following steps are usually performed: collection, drying treatment, cell crushing, and extraction. The disruption of the cell wall mechanically is followed by extraction using organic solvents. Other methods include pressure solvent extraction, supercritical/subcritical fluid extraction, in situ extraction, and two-phase extraction, as shown in Figure 1 [29,30]. People have extensively studied the extraction methods to obtain carotenoids from microalgae and have made some progresses. The yield of β-carotene relied on the total carotenoid mixture obtained and removal of undesired materials.

Figure 2 summarizes the general process of β-carotene extraction from *Dunaliella salina*. It starts by growing *Dunaliella salina*. The growth of *Duncaliella salina* generally happens in facilities that are designed to operate in a closed circuit using seawater or fresh water. The facility can recirculate the culture medium for 10–14 days. Alternatively, this medium can be continuously collected on a daily basis [87]. After that, the biomass of microalgae is harvested and immediately freeze-dried. The dried matters are disintegrated to release the cellular content (steps 1 to 3). Various methods are applied to extract the dried matter, which yield crude extracts containing total carotenoids (step 4), which include β-carotene, canthaxanthin, astaxanthin, lutein, and others, as shown in Table 1 and Table 2. To obtain ingredients enriched in β-carotene or purified β-carotene, contaminants have to be removed. Methods include filtration of the solubilized extract with acetone, direct saponification of the extract with calcium hydroxide to remove chlorophyll, or membrane filtration (steps 5) [88,89,90,91]. The conventional purification method widely used at present is to enrich β-carotene by removing chlorophyll directly with calcium hydroxide via saponification. Some scholars modified it by introducing calcium hydroxide saponification and filtration steps before step 4 (solvent extraction), which saponifies microalgae mass for 2–6 h to remove chlorophyll in an inert gas and at 50–100 °C [29]. Subsequently, β-carotene is extracted after the filtration of saponified residues using halogenated hydrocarbon solvents (e.g., methylene chloride) or hydrophobic solvents (e.g., n-hexane or petroleum ether) and recrystallized in methane chloride/methanol [92]. Using a two-phase bioreactor is a good method recently developed to extract β-carotene from microalgae. It uses biocompatible organic solvent to extract β-carotene specifically without damaging microalgae cells, and to obtain relatively pure β-carotene, directly bypassing the steps that are set for the collection of total carotenoids and saponification (steps 2, 3 and 5) [93,94,95]. A method with advantages of high extraction rate, general, rapid, environmental protection, and low cost is still yet to be developed.

The supercritical carbon dioxide extraction method has been used widely to extract β-carotenes and carotenoids from microalgae such as *Dunaliella salina* [91,96], *Chlorella vulgaris* [91], *Scenedesmus almeriensis* [32], *Synechococcus* sp.[97], *Nannochloropsis* sp. [98], and *Spirulina platensis* [99]. The ratio of 9-*cis*-β-carotene and β-carotene (all-*trans*) is used to predict the antioxidant activity of *Dunaliella salina* extracts [96]. This method has also been used to obtain other compounds such as 25 aroma compounds from sugar cane [100], alkadienes from *Botryococcus braunii* and γ-linolenic acid from *Arthrospira maxima* [91], lipids from *Nannochloropsis* sp. [98]; flavonoids, VA, and α-tocopherol from *Spirulina platensis* [99]; and indolic derivative and PUFAs from *Dunaliella salina* [101]. We acknowledge that this may not be a thorough list of supercritical carbon dioxide extraction methods as we have been focused on the studies of biological activities of β-carotene extracted in this manuscript.

## 5. Studies of Bioactivities of β-Carotene in Microalgae

Carotenoids have a broad spectrum of biological activities. β-carotene has anti-angiogenesis, anti-cancer, and anti-inflammatory effects. Fucoxanthin has anti-angiogenesis, and heart-protection effects [102,103]. As humans cannot synthesize carotenoids, β-carotene has to come from the diet [104]. β-carotene has been recognized by the FDA, the European Community, Japan, the WHO, and other international organizations and experts as a precursor of VA, a food additive, and a nutritional supplement [105]. Naturally derived β-carotene has high biological activity, which can be used in eye diseases, anti-oxidation, anti-aging, cancer prevention, pigmentation in animals, and the enhancement of the animal reproducibility and immune functions [102,106,107]. Streptozotocin-induced diabetic rats have been treated with *Dunaliella salina* extract prepared through pressurized liquid extraction for up to 3 days [108]. The extract without the analysis of the presence of carotenoids appeared to show beneficial effects but did not improve the glucose levels [108].

To summarize the bioactivity of β-carotene from microalgae, β-carotene and microalgae were used as keywords to search the PubMed database and retrieved relevant literature in August 2021. We have limited our search to the studies that have described the methods, analyzed the content for the presence of β-carotene, and tested the biological activities of the biomass, extracts, or purified product and attributed those activities to β-carotene. The search resulted in 255 articles, which include 15 articles that used cells and animals in the experiments. Two of them were irrelevant. The remaining 13 articles include 1 drosophila study [3], 8 rat studies [27,28,109,110,111,112,113,114], 1 cow study [115], 2 human cell line studies [116,117], and 1 antibacterial activity study [118], as summarized in Table 3. The organic solvent extraction is still the main extraction method that is used in 10 articles [27,28,109,110,111,113,114,116,117,118]. HPLC is used in majority of these studies (9/13) to analyze the extract and determine the presence of β-carotene and other carotenoids and bioactive compounds [3,27,109,111,114,115,116,117,118]. One article used supercritical carbon dioxide extraction to obtain carotenoids, identified the presence of β-carotene isoforms, analyzed their biological activities, and determined the underlying functional mechanism [3]. The presence of β-carotenes isoforms is attributed to the extract’s ability to extend the drosophila lifespan [3]. Two articles directly used freeze-drying microalgae powder for research [112,115]. It appears that all these studies only included total carotenoids from microalgae, which the biological activities can be attributed to. Although the presence of β-carotene in the extracts or microalgae mass was confirmed, the purified β-carotene from microalgae has not been used and studied. Therefore, whether the biological activity really can be attributed to β-carotene is still an open question. The synthetic β-carotene is all-*trans*, which is considered less biologically active than the *cis*-β-carotenes [119,120,121]. The naturally extracted β-carotene contains both all-*trans* and *cis* isoforms, which have a higher market prospect and value than the synthetic ones [122,123]. Clearly, future studies using purified β-carotene from microalgae are anticipated and worth being investigated.

In the cell and animal studies, extracts from microalgae extended the lifespan of drosophila via improvement of mitochondrial functions [3], protected the liver functions in rats via the regulation of inflammation and redox status [27,109,110], increased the antioxidative activities in rats [28,111,113,114] and in human dermal fibroblast cells [117], and raised the β-carotene content in cow’s milk [115]. In addition, extracts also being shown to inhibit the growth of pathogenic bacteria on tomato [118]. Some of the studies reported the presence of specific isoforms of β-carotene [3,112], β-carotene [27,110,113,115,116,118], or total carotenoids with or without other components such as PUFA or phenolic acid [27,28,110,111,114,117]. The microalgae sources that β-carotene, total carotenoids, and biomass are derived from are *D. salina* [3,27,109,110,111,112,118]; *S. platensis* [27,28,113,115]; *H. pluvialis* [28]; *B. braunii* [28,114]; *P. lutheri*, *P. palmata*, *P. dioica*, and *C. crispus* [116]; and *Characiopsis aquilonaris*, *Chlorobotrys gloeothece*, and *Chlorobotrys regularis* [117].

## 6. Conclusions and Future Perspectives

As carotenoids have a variety of biological activities, β-carotene is widely used as a bioactive compound in the biomedical field. Currently, β-carotene products on the market are mainly synthetic ones, whereas products extracted from the natural sources only occupy a small fraction. Given the fact that the *cis*-β-carotene isoforms extracted from natural sources are more potent than the synthetic all-*trans*-β-carotene, we can anticipate that microalgae will be used to extract and purify the naturally derived *cis*-β-carotenes on a large scale in the future. To improve the extraction methods, some new technologies such as supercritical carbon dioxide extraction have been developed. The supercritical carbon dioxide extraction method has been used widely to extract carotenoids and other bioactive compounds. It is only a matter of time before more extracts derived from the supercritical carbon dioxide extraction methods will be used in other biological studies using animals or cells in the future. Each extraction method is unique and has a different extraction efficiency. They all have pros and cons. More practical and more effective extraction methods are anticipated in the future.

The extracts of different microalgae sources contain significant amounts of other carotenoids in addition to β-carotene. A variety of methods are applied to show that the extracts contain β-carotene and have positive impacts on animals and cells. These beneficial effects include antioxidant, anti-inflammation, antibacterial, promoting milk production, and increases in lifespan. However, there appears to be a lack of studies using purified β-carotenes to test their bioactivities in cells and animal models. Therefore, whether these positive effects are due to β-carotene or a mixture of bioactive compounds including β-carotene remains to be revealed. In addition, whether β-carotene should be used alone or in combination with other carotenoids to maximize its bioactivities is another open question waiting for answers. Finally, more cellular, animals, and clinical studies are also expected to advance our understanding of the underlying mechanisms by which β-carotene exerts beneficials effects on human and animal health. All these deserve to be explored in the future. Nevertheless, β-carotene derived from microalgae will play an important role in the process.

## Figures and Tables

**Figure 1 foods-11-00502-f001:**
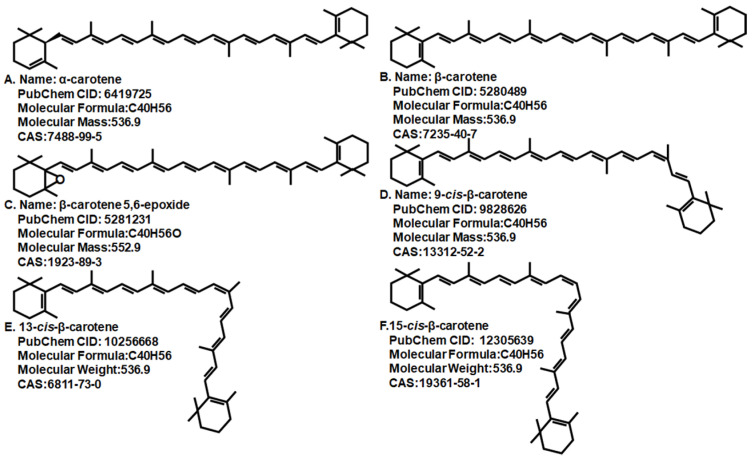
Structures and some physicochemical properties of α-carotene (**A**), β-carotene (**B**), β-carotene 5,6-epoxide (**C**), 9-*cis*-β-carotene (**D**), 9-*cis*-β-carotene (**E**), and 9-*cis*-β-carotene (**F**).

**Figure 2 foods-11-00502-f002:**
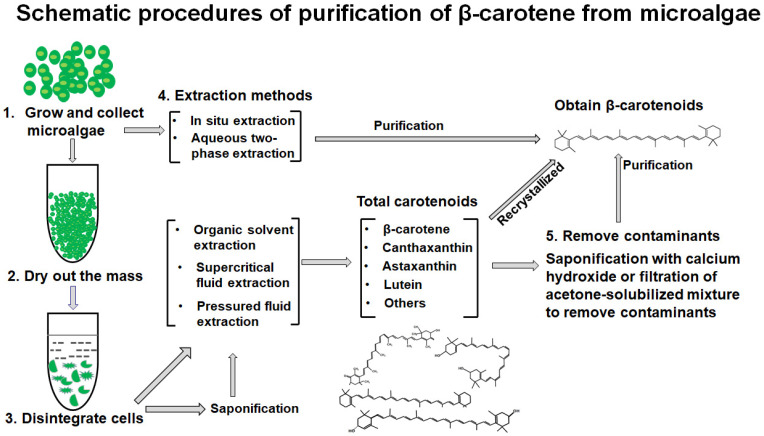
Flow chart of β-carotene extraction from *Dunaliella salina*. Microalgae are grown and collected (step 1), dried (step 2), and disintegrated (step 3). Step 4 is to extract total carotenoids using a variety of methods such as organic solvent extraction, pressurized solvent extraction, or supercritical fluid extraction using the similar compatibility principle of organic solvents. The supercritical fluid extraction method selectively recovers carotenoids by controlling the density of supercritical CO2. Due to the high diffusion-coefficient and low viscosity of supercritical CO2, the extraction time is shorter. Alternatively, in situ or aqueous two-phase extraction can be used to enrich or purify β-carotenoids directly. In addition, saponification and filtration steps can happen before solvent extraction. Step 5 is to remove the chlorophyll component using saponification with calcium hydroxide. After the removal of insoluble matter via filtration, β-carotene is purified using halogenated hydrocarbon solvents (e.g., methylene chloride) or hydrophobic solvents (e.g., n-hexane or petroleum ether). Alternative, the total carotenoids are solubilized in pure acetone and filtered through a 0.45 µm membrane. In situ extraction and two-phase extraction directly extract β-carotene without collecting, drying, or breaking microalgae cells. In the stress condition, β-carotene can be selectively extracted continuously by adding biocompatible organic phases to microalgae.

**Table 1 foods-11-00502-t001:** Contents of α-carotene, β-carotene, β-carotene-5,6-epoxide, 9 or 13 or 15 -cis-β-carotene, and total carotenoids in different microalgae sources.

Strains	Units/Analysis	α-Carotene	β-Carotene	β-Carotene-5,6-epoxide	9, 13 or 15-cis-β-carotene	Total	Ref.
*Chlorella sorokiniana*	µg/g DW/HPLC-PDA-MS for analysis	71	156		42 (9-*cis*)	1408	[75]
*Scenedesmus bijuga*		41	166	13	54 (9-*cis*), 16 (13-*cis*)	1196	
*Chlorella zofingiensis*	mg/g DW/HPLC-APCI-MS/MS for analysis	0.09	0.29				[76]
*Selenastrum bibraianum*		0.08	0.16				
*Desmodesmus denticulatus var. linearis*		0.21	0.4				
*Coelastrum sphaericum*			0.11				
*Mougeotia* sp.			0.14				
*Scenedesmus*	µg/g DW/HPLC–PDA–MS/MS for analysis	42	778	21	124 (9-*cis*), 47 (13-*cis*)	2651	[42]
*Chlorella*			352	49	40 (9-*cis*), 17 (13-*cis*)	1977	
*Aphanothece*			368	62	46 (9-*cis*), 17 (13-*cis*)	1399	
*Nannochloropsis limnetica*	mg/g DW/HPLC chromatograms for analysis		0.28			3.0	[78]
*Mougeotia. salina*		0.084	2.22			5.1	
*Nannochloropsis oceanica*			0.1–1.7				
*Nannochloropsis* sp.			0.67			8.6	
*Nannochloropsis oceanica*			0.3–1.1				
*Nannochloropsis oculata*			0.07–0.14				
*Mougeotia salina*			2.22				
*Nannochloropsis limnetica*			0.28				
*Nannochloropsis* spp.			0.3				
*Porphiridium cruentum*	mg/100 g DW/HPLC chromatograms for analysis		53			167	[79]
*Isochrysis galbana*			53			1760	
*Phaeodactylum tricornutum*			34			1022	
*Tetraselmis suecica*		42	43			297	
*Nannochloropsis gaditana*			100			447	
*Dunaliella tertiolecta*	mg/g DW/HPLC-APCI-MS/MS for analysis	0.04	0.62		0.13 (9-*cis*), 0.06 (13-*cis*), 0.02 (15-cis)	3.4	[77]
*Heterochlorella luteoviridis*		0.47	0.50		0.13 (9-*cis*), 0.12(13-*cis*), 0.04 (15-*cis*)	3.47	
*Eustigmatos magnus*	mg/g DW for total and % of total carotenoids for individuals/HPLC chromatograms for analysis		53			25	[80]
*Eustigmatos polyphem*			51			14	
*Eustigmatos vischeri*			53			19	
*Vischeria helvetica*			58			25	
*Vischeria punctata*			57			33	
*Vischeria stellata*			62			55	
*Chlorella pyrenoidosa*	μg/g DW/HPLC chromatograms for analysis	2466	2155		580 (9-*cis*)		[81]

Note: APCI, atmospheric pressure chemical ionization; DW, dry weight; HPLC, high-performance liquid chromatography; MS, mass spectrometry; PDA, photodiode array; Ref., references.

**Table 2 foods-11-00502-t002:** Contents of carotenoids other than α- and β-carotene in different microalgae sources.

Strains	Unit/Analysis	Total	Lutein	Zea	Anth	Luth	Vio	Vau	Fuco	Ech	Asx	Neo	Crx	Ddx	Dtx	Ctx	Ref.
*Chlorella sorokiniana*	µg/g DW/HPLC-PDA-MS for analysis	1408	909	45			54					131					[75]
*Scenedesmus bijuga*		1196	671	36			33			15		151					
*Chlorella zofingiensis*	mg/g DW/HPLC-APCI-MS/MS for analysis		0.49								5.7		0.11			0.18	[76]
*Selenastrum bibraianum*			1.73								0.41		1.34			0.03	
*Desmodesmus denticulatus var. linearis*			8.46						0.07								
*Coelastrum sphaericum*			2.75								15		0.42			0.21	
*Mougeotia* sp.			1.56						0.92		3.48		0.73				
*Scenedesmus*	µg/g DW/HPLC–PDA–MS/MS for analysis	2651	776	332	38	55	32			273		61	24		21	10	[42]
*Chlorella*		1977	184	271	19	17	13			716		21	15		22	104	
*Aphanothece*		1399	33	103	14					597			10			52	
*Nannochloropsis limnetica*	mg/g DW/HPLC chromatograms for analysis	3.0		0.14	0.34		1.2		0.18			0.42			0.14	0.003	[78]
*Mougeotia salina*		5.1		0.58			1.7		0.01			0.05		0.14		0.14	
*Nannochloropsis* sp.		8.6									6.4						
*Porphiridium cruentum*	mg/100 g DW/HPLC chromatograms for analysis	167		107									6.5				[79]
*Isochrysis galbana*		1760							1643					40	25		
*Phaeodactylum tricornutum*		1022							913					32	44		
*Tetraselmis suecica*		297	85				82										
*Nannochloropsis gaditana*		447		10			337										
*Heterochlorella luteoviridis*	mg/g DW/HPLC-APCI-MS/MS for analysis	3.4	1.8	0.08		0.08	0.59										[77]
*Dunaliella tertio-lecta*		3.5	1.3	0.14			0.79										
*Eustigmatos magnus*	mg/g DW for total and % of total carotenoids for individuals/HPLC chromatograms for analysis	25		2.6	1.9	2.1	15	12									[80]
*Eustigmatos polyphem*		14		3.2	2.2	4.9	2.5	13									
*Eustigmatos vischeri*		2.5		1.9	2.6	3.5	10	14									
*Vischeria helvetica*		25		1.2	2.3	4.2	12	11									
*Vischeria punctata*		33		1.7	2.4	4.7	11	12									
*Vischeria stellata*		55		1.6	1.8	3.6	13	7.6									
*Chlorella pyrenoidosa*	μg/g DW/HPLC chromatograms for analysis		140376	2170			38					259	335				[81]

Note: Anth, antheraxanthin; APCI, atmospheric pressure chemical ionization; Asx, astaxanthin; Ctx, canthaxanthin; Crx, cryptoxanthin; Ddx, diadinoxanthin; Dtx, diatoxanthin; DW, dry weight; Exh, echinenone; Fuco, fucoxanthin; HPLC, high-performance liquid chromatography; Luth, luteoxanthin; MS, mass spectrometry; Neo, neoxanthin; PDA, photodiode array; Ref., references; Vau, vaucheriaxanthin; Vio, violaxanthin; Zea, zeaxanthin.

**Table 3 foods-11-00502-t003:** Microalgae studies that included extraction methods, identified β-carotenes, and determined their biological activities in cells or animal models at the same time.

Strains	Materials/Analysis	Isoforms	Subjects Used	Results	Ref.
*D. salina*	Lyophilized biomass and extracts using supercritical CO_2_ and pre-pared by red light treatment/HPLC (UV-vis, 3D image) chromatograms for analysis	all-*trans* and 9-*cis*-β-carotene	Male and female *Drosophila melanogaster Dahomey*	Extract extends the median lifespan, which is attributed to the improvement of mitochondrial functions by 9-*cis*-β-carotene	[3]
*D. salina*	Pressurized fluid extraction and hexane/HPLC chromatograms for analysis	β-carotene	Bacteria: *P. syringae* pv. *tomato* EPS3, *B. subtilis* ET-1, P. *carotovorum subsp. carotovorum* DSM30168	Hexane extract inhibits bacterial growth, and re-duces speck spot diseases in tomato plants.	[118]
*S. platensis* and *D. salina*	Hexane:isopropyl alcohol (1:1 vol/vol) extraction/HPLC chromatograms for analysis	β-carotene in *S. plantensis*; carotenoids and xanthophyl in *D. salina*	Male and female Wistar rats	The extract of *D. Salina* has better hepato-protective activity than that of *S. plantensis*.	[27]
*D. Salina*	Hexane:ethyl acetate (80:20) extraction/HPLC chromatograms and GC/MS for analysis	β-carotene (15.2% of the algal extract)	Adult male albino Wistar rats	*D. salina* extract protects against TAA-induced hepatic fibrosis in rats.	[109]
*S. platensis*	Spray-dried mass/HPLC for analysis	Diet with 5% spray-dried *S. platensis* that contain β-carotene	Cows	Supplementation of *S. platensis* leads to higher β-carotene content in the milk than the control group.	[115]
*D. Salina*	Hexane: ethyl acetate (80:20) extraction/Repeated chromatographic analysis	Carotenoids	Male Wistar rats	Extracted carotenoids protects age-induced hepatic steatosis via regulating redox status, inflammation, and apoptosis in senescence rats.	[110]
*D. Salina*	n-Hexane: isopropyl alcohol (1:1) extraction/HPLC for analysis	Carotenoids	Wistar rats	Extracted carotenoids have better antioxidant activity than synthetic carotene in rat liver homogenates.	[111]
*D. Salina*	Lyophilized pellets/Absorption spectroscopy at 443 and 475 nm to confirm the presence β-carotene isomers	9-*cis*- and 11-*cis*-β-carotene	Male Wistar rats	9-*cis*-β-carotene in algae pellets might have anti-cancer activity.	[112]
*S. platensis*, *H. pluvialis*, and *B. braunii*	Extracted with acetone, chloroform, methanol, and petroleum ether separately, and pooled/MS for detection	Carotenoids and chlorophyll	Male Wistar rats	The treated rats have higher antioxidant enzymes, and activities in the blood and liver than the controls.	[28]
*P. lutheri*, *P. palmata*, *P. dioica* and *C. crispus*	Extracted with methanol:chloroform 1:1 (*v*/*v*), then water added to collect the organic phase/HPLC chromatograms for analysis	Extracts have 34–42% total fatty acids as n-3 PUFA and 5–7% as pigments, including chlorophyll a, β-carotene and fucoxanthin.	Human THP-1 macrophage cells	Crude extracts inhibit lipopolysaccharide-induced inflammatory responses in human THP-1 macrophage cells.	[116]
*S. platensis*	Lyophilized algae were extracted with ethyl ether and then methanol/Absorption spectroscopy for analysis	Extracts (per liter solvent) have 96.3 mg phenolic, 18 mg tocopherol and 27.5 mg β-carotene.	Male Wistar rats	Extracts show antioxidant activity both in vitro and in vivo.	[113]
*B. braunii* (LB 572)	Acetone extract/HPLC chromatograms for analysis	The extract has 4.7–7.6 µg/mg carotenoids and 11–12.7 µg/mg polyphenols.	Male Wistar rats	The extract acts as antioxidant to reduces free radicals and hydroxy radicals and prevent lipid peroxidation.	[114]
*Characiopsis aquilonaris*, *Chlorobotrys gloeothece* and *Chlorobotrys regularis*	Dichloromethane: methanol (1:1, *v*/*v*) extraction/HPLC chromatograms for analysis	Extracts (mg/g DW) contain respectively chlorophyll a (18.4, 7.3, 17.5), carotenoids (2.3, 1.2, 1.4) and phenolic acid (6.2, 3.2, 5.7)	Normal human dermal fibroblasts	Extracts from these microalgae show antioxidant activities.	[117]

Note: B. braunii, Botryococcusbraunii; C. crispus, Chondrus crispus; DW, dried weight; D. salina, Dunaliella salina; HPLC, high-performance liquid chromatography; GC, gas chromatography; H. pluvialis; Haematococcuspluvialis; P. lutheri, Pavlova lutheri; MS, mass spectrometry; S. platensis, Spirulina platensis; TAA, thioacetamide.

## Data Availability

Not applicable.

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
