# Peer review of "The Extraction of β-Carotene from Microalgae for Testing Their Health Benefits"

_foods, 2022, doi:10.3390/foods11040502_

Round 1
Reviewer 1 Report
It is a revised review article dealing with the extraction of β-carotene from microalgae for testing their health benefits. Although relatively the quality of this review article has improved after revision, there are still some minor issues that need to be addressed.
- Title – “β-carotenoids” should be replace with “β-carotene”
- L59-60 – “as summarized in [6]” should be rewritten as “as summarized in an edited book [6]”
- Table 1, 2 & 3 – the formatting is completely is a mess and should be carefully arranged for all the column and row contents are fully visible within the printable area.
- Table 3 – the botanical names of microalgae provided in the 3rd column are not italicized.
- Figure 1 – A portion of Fig.1F 15-cis-β-carotene structure is missing. Also, the Figure 1 caption is not placed immediately after the Figure 1.
- Section 3 & 4 – the contents of these two sections can be combined under a single section heading.
- Table 1 and 2 – Both “in microalgae” in Table 1 and “in microalgae sources” in Table 2 should be corrected as “in different microalgae sources”.
- L551 – “the appears” should be corrected as “there appears”.
Author Response
Responses to Reviewer #1’s comments
Manuscript NO.: Foods-1592082
Manuscript title: The previous title was " The extraction of β-carotenoids from microalgae for testing their health benefits ".
The authors would like to thank the reviewer #1 for the time and efforts, especially those excellent comments. We have re-revised the manuscript according to the reviewer #1’s comments. Here are our responses to the comments line-by-line. The comments are shown first, which are followed by our responses in italics.
Please note that the words and sentences that have been revised are highlighted in yellow in the main text file. In addition, we have deleted and added references. Therefore, the whole reference list should be considered revised.
Reviewer #1
It is a revised review article dealing with the extraction of β-carotene from microalgae for testing their health benefits. Although relatively the quality of this review article has improved after revision, there are still some minor issues that need to be addressed.
1.1 Title – “β-carotenoids” should be replace with “β-carotene”
Response: Thank you for your suggestion. It has been revised.
1.2 L59-60 – “as summarized in [6]” should be rewritten as “as summarized in an edited book [6]”
Response: Thank you for your suggestion. We have added the words.
1.3 Table 1, 2 & 3 – the formatting is completely is a mess and should be carefully arranged for all the column and row contents are fully visible within the printable area.
Response: Thank you for pointing this out. During the process of trying to fit the information to the table and put the tables into the manuscript template, a lot of things have been changed. Sorry for any confusion. Therefore, we have re-made the tables from the scratch and marked the tables as revised.
1.4 Table 3 – the botanical names of microalgae provided in the 3rd column are not italicized.
Response: Thank you for pointing this out. We have revised them.
1.5 Figure 1 – A portion of Fig.1F 15-cis-β-carotene structure is missing. Also, the Figure 1 caption is not placed immediately after the Figure 1.
Response: Thank you for pointing it out. We have corrected these.
1.6 Section 3 & 4 – the contents of these two sections can be combined under a single section heading.
Response: Thank you for the suggestion. We have combined these into the current section 3. Due the content, we named the previous sections 3 and 4 into sections 3.1 and 3.2, respectively. The numbers of remaining sections are also revised accordingly.
1.7 Table 1 and 2 – Both “in microalgae” in Table 1 and “in microalgae sources” in Table 2 should be corrected as “in different microalgae sources”.
Responses: Thank you for the suggestion. We have revised them accordingly.
1.8 L551 – “the appears” should be corrected as “there appears”
Response: Thank you for pointing it out. Sorry for the mistake. We have revised it.

Reviewer 2 Report
The current format that has been submitted is much cluttered and cannot be evaluated at all in its current form. However, since I had previous versions of this review article, I read and reviewed the figures and tables from it.
The authors have resubmitted a review article on the extraction of β-carotenoids from microalgae for testing their health benefits. It should be of interest to the researchers/scientists and industrial personnel in the field. After careful review, I recommend a major revision and suggest the authors revise this article by addressing the following points:
1-The typographical errors should be double-checked throughout the manuscript. Some examples of these will be mentioned below:
Line 57: … in large scale …. Please change "in" to "on".
Line 60: Microalgae exist in diverse environment conditions … . Please correct this sentence to this form: "Microalgae exist in diverse environmental conditions and are rich sources of biomolecules such as proteins, fat, and carbohydrates."
2- In the introduction, the subject of study is not well presented. You need to talk about beta-carotene, its extraction methods, its health value, as well as the importance of discussing about beta-carotene given the presence of other types of carotenoids in algae.
3- Since there is a lot of information about different types of carotenoids extracted from algae in different parts of the manuscript, it is suggested that the present review article be a review article about the types of carotenoids extracted from microalgae as well as different extraction methods.
4- Please also mention the purification of these carotenoid compounds by chromatographic techniques.
5- The last title should be the conclusion and not the summary! References to the tables in the article are not made in this section.
Author Response
Responses to Reviewer #2’s comments
Manuscript NO.: Foods-1592082
Manuscript title: The previous title was " The extraction of β-carotenoids from microalgae for testing their health benefits ".
The authors would like to thank the reviewer #2 for the time and efforts, especially those excellent comments. We have re-revised the manuscript according to the reviewer #2’s comments. Here are our responses to the comments line-by-line. The comments are shown first, which are followed by our responses in italics.
Please note that the words and sentences that have been revised are highlighted in yellow in the main text file. In addition, we have deleted and added references. Therefore, the whole reference list should be considered revised.
Revierwe#2
The current format that has been submitted is much cluttered and cannot be evaluated at all in its current form. However, since I had previous versions of this review article, I read and reviewed the figures and tables from it.
The authors have resubmitted a review article on the extraction of β-carotenoids from microalgae for testing their health benefits. It should be of interest to the researchers/scientists and industrial personnel in the field. After careful review, I recommend a major revision and suggest the authors revise this article by addressing the following points:
- 1 The typographical errors should be double-checked throughout the manuscript. Some examples of these will be mentioned below:
Line 57: … in large scale …. Please change "in" to "on".
Line 60: Microalgae exist in diverse environment conditions … . Please correct this sentence to this form: "Microalgae exist in diverse environmental conditions and are rich sources of biomolecules such as proteins, fat, and carbohydrates."
Responses: Thank you for pointing them out and your suggestions. Sorry for the mistakes. We have revised the manuscript accordingly.
2.2 In the introduction, the subject of study is not well presented. You need to talk about beta-carotene, its extraction methods, its health value, as well as the importance of discussing about beta-carotene given the presence of other types of carotenoids in algae.
Response: Thank you for pointing this out. Sorry for any confusion. We have revised the corresponding areas according to clarify any confusions.
2.3 Since there is a lot of information about different types of carotenoids extracted from algae in different parts of the manuscript, it is suggested that the present review article be a review article about the types of carotenoids extracted from microalgae as well as different extraction methods.
Response: Thank you for the comments. We have tried to accomplish this task in response to the reviewers’ comments in the previous and this round of revision. This time, we included chromatographic methods in the tables 1 to 3 to enhance this. Please see the revised manuscript for the changes.
2.4 Please also mention the purification of these carotenoid compounds by chromatographic techniques.
Responses: Thank you for your comments. We have added the analytic methods to determine the presence of β-carotene and other carotenoids in tables 1 to 3. Additional comments are also included in the main text. Please see the revised manuscript for the changes.
2.5 The last title should be the conclusion and not the summary! References to the tables in the article are not made in this section.
Response: Thank you for your suggestions. We have revised the title and the languages accordingly. The references to the tables are removed. Please see the revised manuscript for the changes.

Round 2
Reviewer 2 Report
The requested items have been modified. Only one case needs to be revised:
In the conclusion, no reference must be given. Please delete the references given in the conclusion.
Author Response
Manuscript NO.: Foods-1592082
Manuscript title: The extraction of β-carotene from microalgae for testing their health benefits
The authors would like to thank the reviewer #2 for the prompt response. We have re-revised the manuscript according to the reviewer #2’s comments. Here are our responses to the comments line-by-line. The comments are shown first, which are followed by our responses in italics.
Please note that the words and sentences that have been revised are highlighted in yellow in the main text file.
Revierwe#2
The requested items have been modified. Only one case needs to be revised:
In the conclusion, no reference must be given. Please delete the references given in the conclusion.
Responses: Thank you for pointing them out and your suggestions. We have moved the two sentences with citations to the end of the second paragraph of section 5. In addition, we have deleted the citations in the Conclusion section (section 6) and revise the sentence to reflect the changes. Please see the revised manuscript for these changes.
